# Hole doping and electronic correlations in Cr substituted BaFe$_2$As$_2$

Marli dos Reis Cantarino[1,2*], Kevin R. Pakuszewski[3], Bjoern Salzmann[4],
Pedro H. A. Moya[1], Wagner R. da Silva Neto[1], Gabriel S. Freitas[3], Pascoal G. Pagliuso[3],
Cris Adriano[3], Walber H. Brito[5] and Fernando A. Garcia[1,6†]

**1** Instituto de Física, Universidade de São Paulo, 05508-090 São Paulo, SP, Brazil
**2** European Synchrotron Radiation Facility, BP 220, F-38043 Grenoble Cedex, France
**3** Instituto de Física "Gleb Wataghin", UNICAMP, 13083-859, Campinas-SP, Brazil
**4** Département de Physique and Fribourg Center for Nanomaterials, Université de Fribourg,
CH-1700 Fribourg, Switzerland
**5** Departamento de Física, Universidade Federal de Minas Gerais, C.P. 702,
30123-970, Belo Horizonte, MG, Brazil
**6** Ames Laboratory, U.S. DOE, and Department of Physics and Astronomy,
Iowa State University, Ames, Iowa 50011, USA

★ marli.cantarino@esrf.fr ,  † fgarcia@if.usp.br

## Abstract

For a significant composition range, the suppression of the spin density wave transition temperature ($T_{SDW}$) in Cr- and Mn-substituted BaFe$_2$As$_2$ (CrBFA and MnBFA, respectively) coincides as a function of Cr/Mn content, despite the distinct electronic effects of these substitutions. Additionally, for any Cr/Mn content superconductivity (SC) is absent and this topic is particularly less explored in the case of CrBFA. In this work, we employ angle-resolved photoemission spectroscopy (ARPES) and combined density functional theory plus dynamical mean field theory (DFT+DMFT) to address the evolution of the Fermi surface (FS) and electronic correlations in CrBFA. Our findings reveal that incorporating Cr leads to an effective hole doping of the states near the FS, which is well described within the virtual crystal approximation (VCA). Moreover, analysis of the ARPES spectra of the bands with main $d_{yz}$-orbital character reveals a fractional scaling of the imaginary part of self-energy as a function of the binding energy, a signature property of Hund's correlations. Our DFT+DMFT calculations support these experimental findings. We conclude that CrBFA is a correlated electron system for which the changes in the FS as a function of Cr are unrelated to the suppression of $T_{SDW}$. In addition, we suggest that the absence of SC is primarily due to the competition between Cr local moments and the Fe-derived itinerant spin fluctuations.

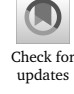

# 1  Introduction

The discovery of superconductivity (SC) in hole-doped $LaO_{1-x}F_x FeAs$ [1] led to the new field of the iron-based superconductors (FeSCs). A superconducting critical temperature ($T_{SC}$) as high as $\approx 55$ K in hole-doped $SmO_{1-x}F_x FeAs$ was soon after reported [2] and remains to date among the highest so far observed in this diverse family of materials [3].

The $BaFe_2As_2$ (BFA) material is a particularly well-explored parent compound of the FeSCs. It undergoes an antiferromagnetic (spin density wave, SDW) transition with a critical temperature ($T_{SDW}$) of $\approx 133.7$ K that is preempted by an almost concomitant tetragonal to orthorhombic phase transition [4,5]. In BFA, partial chemical substitutions on the Ba, Fe or As sites can stabilize high-temperature superconductivity (HTSC) [6–11]. The highest $T_{SC}$, $\approx 38$ K, is observed when Ba is partially substituted by an Alkaline metal, as in $Ba_{1-x} A_x Fe_2As_2$ ($A =$ K or Cs) [12,13], corresponding to nominal hole doping.

In all cases wherein HTSC emerges in BFA substituted phases, the composition ($x$) $vs.$ temperature ($T$) phase diagram shows that the maximum $T_{SC}$ is observed in a $x$ range wherein the $T_{SDW}$ is fully suppressed. This phenomenology suggested a close link between fluctuations of the magnetically ordered phase and the formation of HTSC, which is part of the consensus in this research field [14]. The relevant energy scales to understand the FeSCs, however, remain under debate.

Low energy effective models contain some essential ingredients to understand the phase diagrams of many FeSCs [15,16]. This scenario posits that the SDW phase is the result of a nested Fermi surface (FS) and that charge doping detunes the nesting condition, thus suppressing $T_{SDW}$. The resulting fluctuations boost the SC pairing. Other descriptions adopt the $Fe^{2+}$ $3d^6$ local electronic structure as a starting point. The resulting large spins are lowered by kinetic frustration and the main energy scales are provided by the on-site electron-electron Hund's exchange and by the Coulomb interactions [17–20]. Within the latter framework, the FeSCs are classified as Hund's metals, a new class of strongly correlated materials, where the strength of correlations are sensitive to the Fe-$3d$ occupancy, Hund's coupling $J_H$, and to the pnictogen/chalcogen height. More importantly, the Hund's metals exhibit an orbital(charge) and spin separation for a broad intermediate temperature region, where the orbital(charge) are itinerant and the spin degrees of freedom are quasi-localized [21]. The resulting HTSC phase from this electronic correlated state displays an universal heat capacity associated with $2\Delta_{max}/T_{SC} = 7.2 \pm 1$ (where $\Delta_{max}$ is the maximum SC gap) which can be explained by taking into account the incoherent nature of local spin fluctuations [22].

To distinguish between possible scenarios, it is key to probe how chemical substitutions change the electronic structure of the FeSCs. In this context, an intriguing asymmetry is observed in the phase diagram of transition metal substituted BFA: SC is not observed in the Cr [23], Mn [24] or V [25] substituted materials, which corresponds to the nominal hole doping. Particular attention has been devoted to the Mn and, to a lesser degree, Cr substituted materials (hereafter called, respectively, MnBFA and CrBFA).

In the case of MnBFA, the absence of SC was first ascribed to the lack of charge doping by Mn [26,27]. A more complete scenario, however, encompasses the scattering between Mn- and Fe-derived magnetic excitations [28, 29] as well as electronic disorder and correlations [30], which is in line with theoretical calculations [31–33].

It was soon noted that the CrBFA and MnBFA $x$ vs. $T$ phase diagrams look very similar, with the suppression of $T_{SDW}$ depending only on $x$ [24]. Indeed, both Cr and Mn induce a crossover from an itinerant to a more localized form of magnetism [34,35] and, as in MnBFA, it is suggested that the Fe-SDW and Cr-Néel fluctuations compete for the ground state [23, 36–38]. Similar effects are caused by Cr-substitutions in Ni-doped BFA [39–41] as well as in P-substituted BFA [42]. Disordered magnetism, however, is only observed in MnBFA [43]. In addition, the role of Cr as a hole dopant is observed in Ni-doped BFA [39] but recent angle-resolved photoemission spectroscopy (ARPES) experiments of Cr-substituted $CsFe_2As_2$ do not support hole doping caused by Cr [44]. All this phenomenology suggests that, as in the MnBFA phase diagram, the Cr effects on the electronic structure and correlations in CrBFA require clarification.

This work is dedicated to understanding hole doping and electronic correlations in CrBFA. We employed ARPES experiments and density functional theory in combination with dynamical mean-field theory (DFT+DMFT) calculations of the excitation spectra of $Ba(Fe_{1-x}Cr_x)_2As_2$ ($x = 0.0$, 0.03 and 0.085, hereafter called the BFA, Cr3% and Cr8.5% materials, respectively). Our results show that Cr is an effective hole dopant, as in the case of K substitution [45], with the DFT+DMFT calculations capturing the experimentally observed changes in size and shape of the hole and electron pockets.

Moreover, based on the experimental ARPES spectral function, we analyze electronic correlations in CrBFA. We focus on electronic bands with main $d_{yz}$-orbital character which possesses clear spectroscopic features. We find that the imaginary part of the self-energy, $\text{Im}\Sigma(E_B)$, presents a Cr-dependent fractional scaling as a function of the binding energy ($E_B$), the hallmark of a Hund's metal [46]. Our calculations also reproduce this feature, including the Cr dependency of the scaling fractional exponent.

## 2  Materials and methods

Single crystals of $Ba(Fe_{1-x}Cr_x)_2As_2$ were synthesized by an In-flux method [47]. The resulting crystals were crushed and sieved into a fine powder for X-ray diffraction (XRD) experiments to check the crystallographic phase and determine lattice parameters. The final Cr content was checked by energy-dispersive x-ray spectroscopy (EDS) measurements. Physical properties (resistivity and specific heat) were characterized by a commercial Physical Properties Measurement System (PPMS) from Quantum Design. Results from EDS and physical properties were compared to composition vs. $T$ phase diagrams in literature [23, 35, 36] to benchmark the values of $x$.

The samples with $x = 0.0$, $x = 0.03$, and $x = 0.085$ were selected for ARPES measurements at the Bloch beamline of the Max IV synchrotron in Lund, Sweden. The ARPES spectra were obtained using the Scienta DA30 photoelectron analyzer for incident photon energies between 60 and 81 eV. At this photon energy, an energy resolution of about 8 to 10 meV and angular resolution of 0.1° was achieved. The samples were glued on a Mo sample holder using silver epoxy. An Al post was glued at the top of each sample for subsequent cleaving. The post was removed inside the main preparation chamber (vacuum of $3 \times 10^{-10}$ mbar). The samples were then transferred to the analyzer chamber, at $2 \times 10^{-11}$ mbar.

The Brillouin zone (BZ) high-symmetry directions are labeled according to the body-centered tetragonal crystal structure. The results presented were measured at the samples' tetragonal PM state ($T = 150$ K). During experiments, we probe the high-symmetry directions $\Gamma X$ and $\Gamma M$ for both $\Gamma$ and $Z$ $k_z$ levels. We employed linear horizontal (LH) and vertical (LV) polarization to probe different Fe-3$d$ orbital contributions to the ARPES intensity.

The DFT+DMFT calculations were performed using the fully charge self-consistent DFT + embedded-DMFT approximation [48] at 150 K. The DFT calculations were performed within the full potential linearized augmented plane wave method and Perdew-Burke-Ernzehof generalized gradient approximation (PBE-GGA) [49], as implemented in WIEN2k package [50]. The DMFT impurity problem was solved by using continuous-time quantum Monte Carlo (CTQMC) calculations [51], and rotationally invariant interaction with $U = 5.0$ eV and Hund's coupling $J = 0.8$ eV. Similar U and J values were successfully employed in Ref. [18] within the same implementation.

To calculate the spectral functions and Fermi surfaces we performed the analytical continuation of the calculated self-energies using the maximum entropy method [48]. For the double-counting correction term, we used the standard fully localized-limit form [52] with nominal occupancy $n_d^0 = 6$. We used the experimental crystal structures obtained by XRD and the Cr doping was simulated within the virtual crystal approximation (VCA). The effect and validity of VCA is further discussed in the Appendix A.

## 3 Results and discussion

An overview of the experimentally determined electronic band structures of CrBFA is presented in Fig. 1($a$)-($c$). The experimental geometries, beam polarization (either LH or LV polarizations), and the sample's Cr content are indicated in each panel. For all samples and measurement conditions, the band features are well observed allowing their characterization as a function of Cr.

The band positions are marked by the colored dots determined from the second derivatives of the band maps and from Momentum and Energy Distribution Curves (MDCs and EDCs, respectively). Indeed, having the spectral function a Lorentzian lineshape, its central position is the minimum of the second derivatives. This method, therefore, describes the band shape and effective mass to a point but ignores information about the one-particle excitation lifetime and scattering rate.

For this family of materials, the electronic bands derive from the Fe 3$d$-states and are subjected to the As ligand effects, which breaks the Fe 3$d$-states degeneracy and imparts a strong orbital character to the electronic bands [3,14]. In Fig. 1($a$)-($c$), the distinct main orbital characters are labeled by dots of different colors and were determined by the selection rules for the ARPES intensity polarization dependence and guided by previous works [53–56].

The Cr hole doping effect on the bands forming the hole pockets around $\Gamma$ is visible from direct inspection of the data. Indeed, the hole pocket Fermi vectors, $k_F$, are increasing with Cr introduction, in line with the expectation due to hole-doping. Even for the highest Cr doping level in this work (Cr8.5%), a total of three hole pockets are still visible around $\Gamma$, being mainly observed in pairs because of the polarization selection rules.

In the cases of BFA and Cr3% samples shown in Fig. 1(a) and (b), the bands straightforwardly follow the selection rules. Indeed, around the $\Gamma$ point, for measurements along the $\Gamma X$ direction and adopting LH polarization (upper left panels in the figures), one can distinguish the two bands with $d_{xz/yz}$ that form the inner hole pockets. The band forming the outer hole pocket is absent in this configuration but is readily observed in the right upper panels of the same figures, which present measurements along the $\Gamma M$ direction adopting LH polarization.

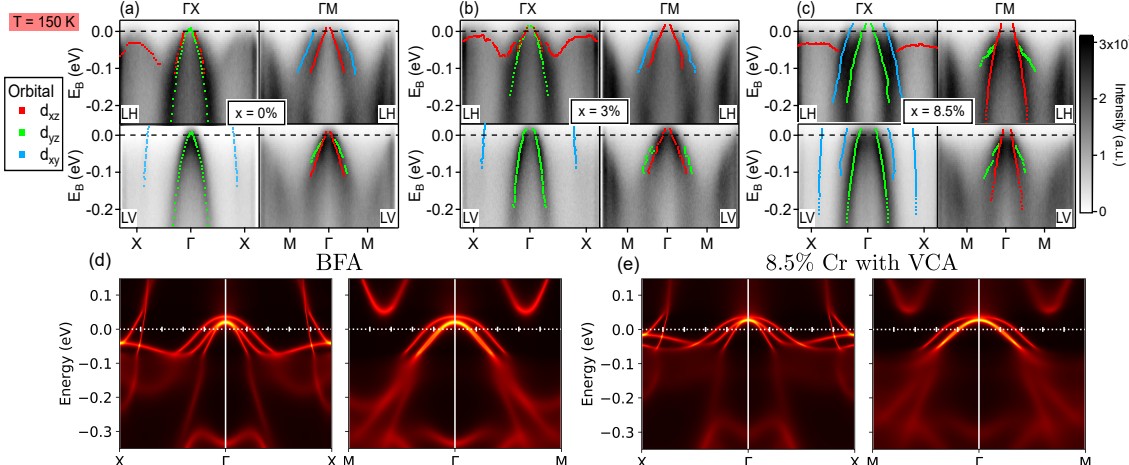

Figure 1: $(a)$-$(c)$ Overview of the ARPES measured electronic band structure of the BFA, Cr3% and Cr8.5% materials. As indicated, measurements were taken along the $\Gamma X$ and $\Gamma M$ directions and for LH and LV polarizations. The dots represent the band positions as obtained from the second derivative of the band maps and MDC analysis. DFT+DMFT spectral functions for the paramagnetic phases at $T = 150$ K for $(d)$ BFA and $(e)$ Cr8.5% in the VCA.

The situation with the Cr8.5% sample is more involved. Closer inspection reveals a change in the polarization selection rules at this substitution level. In the upper left panel of Fig. 1(c), the spectral weight of the outer hole pocket is seen crossing the Fermi surface in an experimental configuration ($\Gamma X$, LH polarization) for which one expects to observe only bands with $d_{xz/yz}$ main orbital character. This is evidence that the main orbital character of the $d_{xy}$ derived bands and their hybridization are significantly affected by Cr, as observed for Co substitution [57]. In addition, comparing the upper and bottom right panels of Fig. 1(c), one observes a relatively weak polarization dependence. Empirically, it seems that the same bands are observed for the two distinct polarizations.

Based on the results for the BFA and Cr3% samples, we can pinpoint that the band with $d_{yz}$ orbital character (green band) contributes to the spectrum in the lower right panel for the Cr8.5% sample ($\Gamma M$, LV polarization) but we observe that on the upper right panel this same band seems also present, albeit in an experimental configuration where one would expect to observe the outer hole pocket. This also evidences the underlying change in the hybridization of the bands with main $d_{xz/yz}$ orbital characters.

The changes in hybridization may result from the structural changes caused by Cr substitutions. It may affect the bands even if charge doping is weak, as concluded from experiments of MnBFA samples [30, 58]. To draw a more quantitative understanding of the band evolution with doping and to assist the band assignment, we show DFT+DMFT calculations of the BFA (Fig. 1($d$)) and Cr8.5% (Figs. 1($e$)) spectral functions. These calculations were performed by freezing the structure as that of BFA and considering charge doping effects within the virtual crystal approximation (VCA). The validity and limitations of this approach are discussed in Appendix A.

Comparing calculations and experiments, one can confirm the presence of three bands forming hole pockets around the $\Gamma$ point and the expansion of these pockets can also be inferred from the calculations and crosschecked with experiments. On the other hand, it is intriguing to observe that the corresponding change in the size of electron pockets due to doping is not readily observed along the $\Gamma X$ direction in both experimental and calculated band maps. To clarify this topic, we resort to a direct inspection of the FSs.

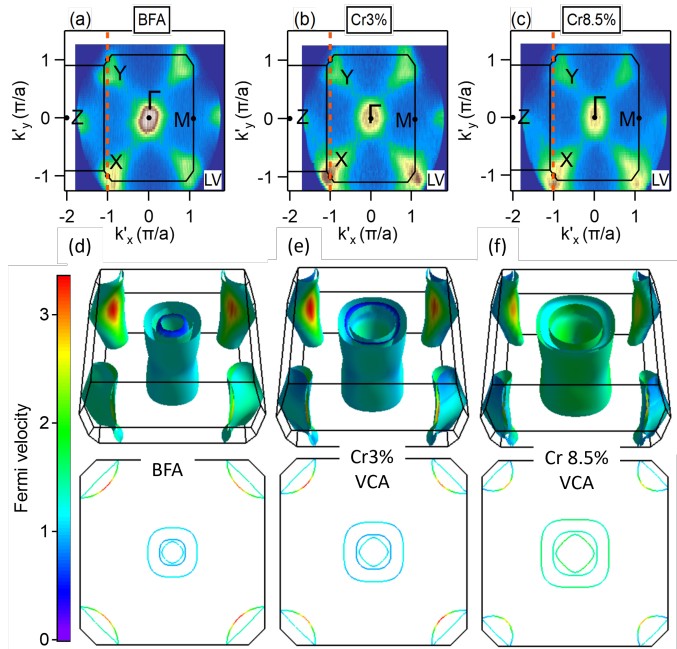

Figure 2: $(a)-(c)$ Measured Fermi Surface of the BFA, Cr3% and Cr8.5% materials with LV polarization, showing the BZ draw and its high-symmetry points. The red dashed line indicates the $XY$ cut based upon which the electron pockets of 3$(h)$ were reconstructed. $(d)-(f)$ DFT+DMFT paramagnetic ($T = 150$ K) Fermi Surface for BFA, Cr3% and Cr8.5%.

In Figs. 2$(a)$-$(c)$ the experimentally obtained FSs, for the three samples are presented. The data suggest that the electron pockets are changing, evolving from an idealized elliptical shape in BFA to a more petal-like shape in CrBFA. It also suggests that the pockets are shrinking in a direction other than the $\Gamma X(Y)$ direction. To guide our analysis, we present in Figs. 2$(e)$-$(f)$ the corresponding DFT+DMFT calculations of the three-dimensional FSs of our materials, along with cuts of the $\Gamma$-centered FSs. These FSs were obtained by considering vanishing scattering rates at the chemical potential.

The calculations show all three hole pockets increasing upon Cr introduction, but the decrease of the electron pockets area is not as clear. The calculated electron pockets for the doped samples show a protuberant shape along the $\Gamma X$ direction and seem to decrease along the $XY$ direction. This motivates the exploration of different high-symmetry cuts of the experimentally obtained electron pocket bands as a function of Cr content. From the ARPES band map measurements, it is possible to extract the electronic band as a function of $k_{y(x)}$ and $E_B$ for other high-symmetry directions by fixing the map $k_{x(y)}$ to a high-symmetry point and reconstructing the energy bands.

The $YZ$ and $XY$ cuts can be adopted to capture the changes in electron pockets as a function of Cr. These cuts are represented as green and magenta dashed lines #1 and #2 in Figs. 3$(a)$-$(b)$ for the Cr8.5% sample. The red dashed lines in Figs. 2$(a)$-$(c)$ represent the $XY$ cut for all three samples. The full set of electronic band positions as a function of Cr content is compiled in Figs. 3$(c)$-$(h)$, of which $(f)$ and $(h)$ present band positions extracted from the mentioned $YZ$ and $XY$ cuts, respectively. The measurement conditions and main orbital characters attributed to each band are labeled on each figure, where the darker points represent the parent compound and lighter points represent a larger Cr content. The electron pocket along the YZ direction has different selection rules since it is related to the inner section of the electron pocket, also interpreted as the minor axis of an idealized ellipse.

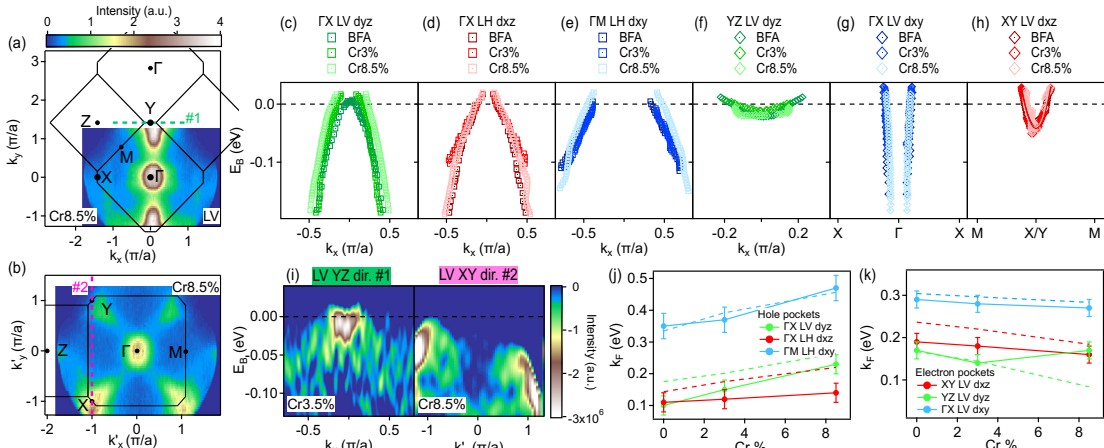

Figure 3: (a) and (b) experimentally determined FSs of the Cr8.5% sample with LV polarization at two different geometries. The BZ and high-symmetry points are indicated for reference and the colored dashed lines (green and magenta) are guides to eyes indicating the FS cuts #1 and #2 from which the electron pockets presented in (f) and (h) were reconstructed. (c)−(f) Survey of the band state positions, in the vicinity of $E_F$, obtained from the fitting of the second derivatives of the MDCs for different Cr content. In each panel, experimental conditions are indicated. (i) YZ and XY second derivative energy maps reconstructed from cut #1 and #2 for the Cr3.5% and Cr8.5% samples respectively. (j) and (k) Evolution of Fermi vectors ($k_F$) as a function of Cr content for hole pockets and electron pockets. The dashed lines represent the $k_F$ values predicted by our calculations.

More specifically, in Figs. 3(c)-(e), the Cr-dependent evolution of the hole pockets is presented. The tendency of increasing hole pockets is clear for all bands as is indeed observable by direct inspection of the band measurements (Fig. 1). Figs. 3(f)-(h) present the electron pockets evolution. The measurements along $\Gamma X$ in Fig. 3(g) show a weak Cr-dependency. In the case of Figs. 3(f) and (h), the reconstructed bands from the Fermi maps have lower resolution, due to the electron analyzer collection mode, and even with second derivative analysis the data are not as well-defined as those obtained from the band maps. This is illustrated in the two panels of Fig. 3(i), wherein representative second derivative data of this band reconstruction are presented.

To better visualize the band evolution, we can study the Fermi vectors $k_F$ as a function of Cr content and compare the results with the theoretically calculated values. The Fermi vectors are the momentum points where a certain band crosses the Fermi level, characterizing the pocket sizes and their respective increasing/decreasing associated with effective charge doping. In Fig. 3(j), the filled circles show the experimental $k_F$ values of the hole bands, and the dashed lines represent the theoretical results, extracted from the Fermi surfaces of Figs. 2(d)-(f). The experimental data seem to follow the predicted trend, with excellent agreement observed in the case of the outer hole pocket of $d_{xy}$ main character. Due to the multiband nature of the electronic structure, it is hard to disentangle the contributions of the inner and middle hole pockets of $d_{xz/yz}$ main orbital character, and the agreement is not as good. Nevertheless, the trend stands and we can affirm that the VCA calculations correctly describe the effective hole doping and hole pocket size increasing.

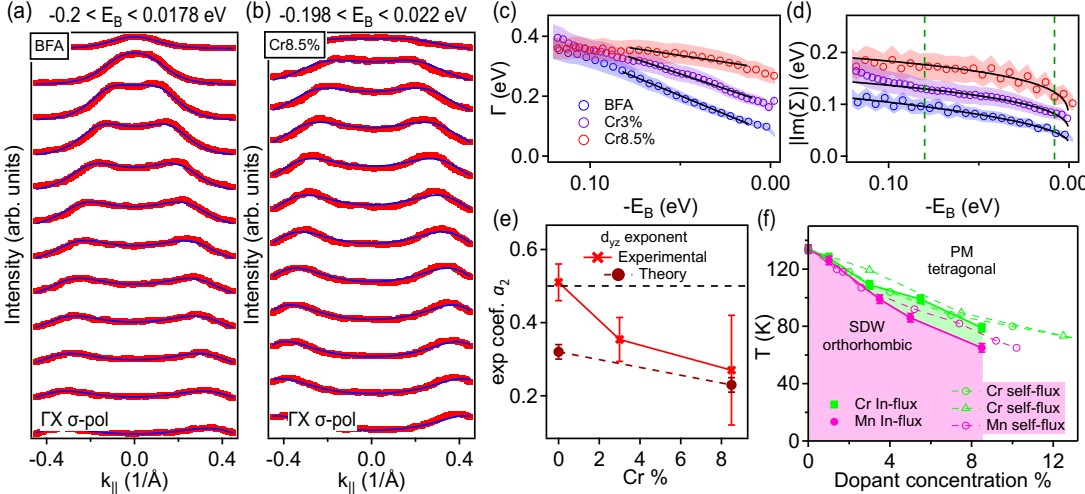

Figure 4: (a)-(b) ARPES spectral function analysis for the electronic band with main $d_{yz}$-orbital character showing fittings (blue lines) of several MDCs (red dots) for increasing binding energies. The extracted quantities (c) $\Gamma(E_B)$ and (d) $\text{Im}\Sigma(E_B)$. The shaded areas represent the error bars and the solid black lines are fittings close to $E_F$ that are linear for $\Gamma(E_B)$ and a power-law for $\text{Im}\Sigma(E_B)$. (e) The fitted exponents of (d) as a function of Cr content compared with the calculated ones (see Table 1). (f) $T$ vs. $x$ phase diagram comparing CrBFA samples used in this study with MnBFA samples grown by the In-flux method. The results for the self-flux grown samples are from references [23, 24, 35]. Care was taken to compare $T_{\text{SDW}}$ values obtained by the same method.

As for the electron pockets, the predicted decrease in the pocket size is small and, experimentally, the variation of $k_F$ could be considered constant within error bars, as shown in 3(k). For the electron pocket extracted from the high-statistic electron bands along the $\Gamma X$ direction ($d_{xy}$ orbital character), one can observe that the experimental data is in excellent agreement with the theoretical dashed lines: we see a small decrease in the size of the electron pockets. For the reconstructed bands, as explained, the statistics are not as good, but we can still see a decreasing trend in the data, which closely follows the dashed theoretical lines. The exception is the result obtained for a cut along the YZ direction in the case of the Cr8.5% sample. Nonetheless, we can confirm that the electron pockets are slightly decreasing, a tendency relatively well described by our theoretical predictions.

Having established Cr as a hole dopant in BFA, we now focus on the electronic correlations. We seek to fit momentum distribution curves (MDCs) to the expression for the one-particle spectral function $A(\boldsymbol{k}, E_B)$ for a system of correlated electrons [59]. The goal is to extract the scattering rate, $\Gamma(E_B)$, and the imaginary part of the self-energy, $\text{Im}\Sigma(E_B)$, from the MDCs analysis for the band with $d_{yz}$ main orbital character. We adopt the data obtained in measurements along $\Gamma X$ and LV polarization (represented in the left lower panels in Fig. 1($a-c$) as the green bands forming the hole pockets). In this configuration, this band is well-separated from other spectral contributions, providing a clear spectra set for the fitting. The analysis is carried out as in Refs. [60, 61] and the positions presented in Fig. 1($a-c$) are adopted as initial guesses in the fitting.

Table 1: Self-energy imaginary part $\text{Im}\Sigma(E_B)$ fitting coefficients obtained from the theoretical calculations and experimental data for the bands derived from the $d_{yz}$ orbital. The fitted function is $|\text{Im}\Sigma(E_B)| = a_0 + a_1(-E_B)^{a_2}$.

| Orbital | Fitting | BFA | | | Cr8.5% | | |
|---|---|---|---|---|---|---|---|
| | | $a_0$ | $a_1$ | $a_2$ | $a_0$ | $a_1$ | $a_2$ |
| $d_{z^2}$ | theory | -0.19(4) | 0.82(4) | 0.42(3) | -0.27(6) | 0.97(6) | 0.35(3) |
| $d_{x^2-y^2}$ | theory | -0.15(3) | 0.74(3) | 0.44(2) | -0.21(4) | 0.85(4) | 0.38(2) |
| $d_{xz}$ | theory | -0.29(4) | 0.95(4) | 0.32(2) | -0.52(6) | 1.26(6) | 0.22(1) |
| $d_{yz}$ | theory | -0.27(4) | 0.93(4) | 0.32(2) | -0.47(7) | 1.21(7) | 0.23(2) |
| | exp. | 0.02(1) | 0.26(6) | 0.50(5) | 0.05(2) | 0.23(6) | 0.27(15) |
| $d_{xy}$ | theory | -0.41(7) | 1.08(7) | 0.26(2) | -1.34(31) | 2.11(32) | 0.12(2) |

In Fig. 4(a)-(b), we present these fittings for the BFA and Cr8.5% samples. A clear necessary caution with this type of analysis is that substitutional disorder contributes with extrinsic effects to the broadening of the spectroscopic features. In turn, it poses a challenge to the determination of $\Gamma(E_B)$ and thus of $\text{Im}\Sigma(E_B)$. To evade this problem, we focus on the scaling properties of $\Gamma(E_B)$ and $\text{Im}\Sigma(E_B)$, which already contain all qualitative information about the correlated nature of the electronic states in our samples, and are robust against the homogeneous broadening due to disorder. With this in mind, we can analyze the extracted values of $\Gamma(E_B)$ and the calculated $\text{Im}\Sigma(E_B)$ as a function of $E_B$, shown in Figs. 4(c) and (d) for all samples. The shaded area represents the error bars.

It is clear that $\Gamma(E_B)$ and $\text{Im}\Sigma(E_B)$ are not proportional to each other and do not follow a quadratic behavior, which would be expected for a normal Fermi liquid. This is an indication of the correlated nature of the metallic state in BFA and CrBFA and was also observed for other substitutions [30, 60–62]. In the case of the scattering rates, this is qualitatively captured by the lines drawn in Fig. 4(c), which suggest that $\Gamma(E_B)$ can be described as a linear function of $E_B$ close to $E_F$.

Most interesting, the $\text{Im}\Sigma(E_B)$ for all samples present a fractional scaling close to $\sqrt{-E_B}$, the hallmark of a Hund's metal [46, 63]. The data is presented in Fig. 4(d) and the fractional behavior is shown as a fitting to the function $|\text{Im}\Sigma(E_B)| = a_0 + a_1(-E_B)^{a_2}$, represented by the black solid lines. Only points between the two green dashed lines were considered for the fitting, excluding the points close to the $E_F$ within the binding energy resolution and points too far from the FS.

A similar fitting was performed for the DFT+DMFT obtained self-energies on the Matsubara axis (imaginary axis) for the BFA and Cr8.5% samples. Results of the analysis for the theoretical data are presented in Table 1 for all bands, along with the experimental result for the band with main $d_{yz}$ orbital character.

The $a_0$ and $a_1$ coefficients determined from experiments do not follow a particular trend, which can be expected since they are closely related to the disorder and the self-energy extraction method. The power-law scaling, however, which is related to the $a_2$ coefficient, has a robust trend. For comparison, theoretical and experimental results are compiled in Fig. 4(e). The experiments and theory compare well, with $a_2$ tending to decrease as a function of Cr concentration. The dashed black line is drawn to pinpoint the value of 0.5 corresponding to a square root dependency of $\text{Im}\Sigma(E_B)$ with $E_B$.

Electronic correlations can also be examined by extracting the quasiparticle weight, or renormalization factor, (Z). From our calculations, Z can be extracted from the self-energy without using the parabolic band dispersion approximation. These calculations show Z values between $0.3 - 0.5$, with Z decreasing (more correlated) with doping. From experiments, the mass renormalization ($m^*$) can be obtained by parabolic fittings of the band dispersions. We observed that near the Fermi surface the outer hole pocket, of $d_{xy}$ orbital character, is much heavier ($m^*/m_e = 15$) than predicted by theory ($m^*/m_e = 2.7$) and that the orbital dependence of the renormalization factors is stronger. These results, however, may only indicate that accessing correlations by the parabolic approximation is not the best strategy in the case of Hund's metal [64].

In Fig. 4(f) we show the phase diagrams for the CrBFA samples used in this work and compare them with other CrBFA and MnBFA samples. We emphasize that our phase diagram for In-flux grown samples is in agreement with those of self-flux grown samples [23,24,35]. By inspecting the phase diagrams, it seems fair to conclude that the evolution of $T_{SDW}$ in the cases of MnBFA and CrBFA is not strongly dependent on the nature of the substituent atom, despite the rather distinct effects of Cr and Mn on the low energy degrees of freedom. Indeed, Cr is a hole dopant whereas Mn is not. Thus, the electronic bands tuning near $E_F$, by either Cr or Mn, is not controlling $T_{SDW}$. Rather, we suggest that the competing SDW and Néel phases derived, respectively, from the Fe and Cr/Mn spins is the main control parameter. This mechanism only depends on the total amount of extrinsic spins introduced in the Fe lattice, and therefore only depends on $x$.

## 4 Summary and conclusions

The main result of our paper is that the partial substitution of Fe by Cr causes an effective hole doping of the electronic states in the vicinity of the FS. This is supported by direct inspection of the data and, in more details, by the agreement between our experiments and calculations. We then proceed to discuss results based on our analysis of the ARPES spectral functions obtained for the band with $d_{yz}$ orbital character. We found that the imaginary part of the self-energy, $\mathrm{Im}\Sigma(E_B)$, presents a Cr-dependent fractional scaling as a function of the binding energy, a sign of Hund's correlations. This scaling behavior was also observed in our DFT+DMFT calculations.

By comparing $x$ vs. $T$ phase diagrams for CrBFA and MnBFA, we concluded that the suppression of $T_{SDW}$ cannot depend on the effects caused by Cr and Mn on the Fermi surfaces. A recent analysis of Mn and Cr substituted 1144 FeSCs materials [65,66], also suggests that the amount of doped holes is not controlling the suppression of $T_C$ and $T_{SDW}$ for Cr and Mn substitutions, making the breakdown of FS nesting scenario possibly more ubiquitous. The characterization of CrBFA and MnBFA as a Hund's metal naturally explains these results.

Recently, it was suggested that the Cr doping of $CsFe_2As_2$ could push the orbital-selective Hund's metal system further into a strongly correlated electronic phase reminiscent of Heavy Fermion quantum criticality [67]. This agrees with our findings in the low Cr-doping region, where we observed the increase in electronic correlations induced by Cr.

The absence of SC in Cr- and Mn-substituted $BaFe_2As_2$ remains a topic of discussion despite intensive research in the past years. In the case of MnBFA, recent experimental work proposes that electronic correlations and the disordered scattering of the Fe-derived magnetic excitations should be part of any minimal model addressing this problem [29,30]. In the case of CrBFA, the present paper shows that what distinguishes Cr from another hole-doped material, as K-substituted BFA, is indeed the underlying competition between the Cr- and Fe-derived magnetism. This finding suggests that the magnetic scattering of the Fe-derived excitation by Cr suppresses SC in CrBFA.

## Acknowledgments

We thank Claude Monney for fruitful discussions. We acknowledge the MAX IV Laboratory for beamtime on the Bloch Beamline under Proposal 20200293 and the support of Gerardina Carbone and Craig Polley during experiments.

**Author contributions**   F.A.G. and M.R.C. designed the project. K.R., G.S.F., P.G.P, and C.A. were responsible for the single crystal growth and characterization. M.R.C., B.S., P.H.A.M., and F.A.G. performed the ARPES experiments. M.R.C. conducted the ARPES data processing and analysis. W.H.B. performed the DFT calculations. The draft was written by M.R.C., W.H.B. and F.A.G. with inputs from all co-authors.

**Funding information**   The Fundação de Amparo à Pesquisa do Estado de São Paulo financial support is acknowledged by M.R.C. (Grants No. 2019/05150-7 and No. 2020/13701-0), F.A.G. (Grant No. 2019/25665-1); K.R.P., P.G.P. and C.A. (Grant No. 2017/10581-1). B.S. acknowledges funding from the Swiss National Science Foundation (SNSF) Grant No. P00P2_170597. P.G.P. and C.A. acknowledge financial support from CNPq: Grants No. 304496/2017-0, 310373/2019-0, and 311783/2021-0. W.H.B acknowledges the financial support from CNPq Grant No. 402919/2021-1, FAPEMIG, and the computational centers: National Laboratory for Scientific Computing (LNCC/MCTI, Brazil), for providing HPC resources of the SDumont supercomputer (http://sdumont.lncc.br), and CENAPAD-SP. F. A. G. was partially supported by the U.S. Department of Energy, Office of Science, Basic Energy Sciences, Materials Science and Engineering. Ames National Laboratory is operated for the USDOE by Iowa State University under Contract No. DE-AC02-07CH11358.

## A   Note on the theoretical calculations

As discussed in the main text, the difference between the structural and charge doping effects introduced by Cr is not evidenced in the experiments. We discussed DFT+DMFT calculations of the spectral functions of pristine BFA (Fig. 1(d)) and of Cr8.5% (Figs. 1(e)) within the virtual crystal approximation (VCA). To deepen our understanding, we adopted another strategy to capture the effects caused by Cr: we perform calculations considering the Cr8.5% crystal structure, without the VCA. In this case, we probe only how the changes in the crystal structure, and the change in electronic hybridization that follows from it, affect the electronic structure. This is shown in Fig. 5(a).

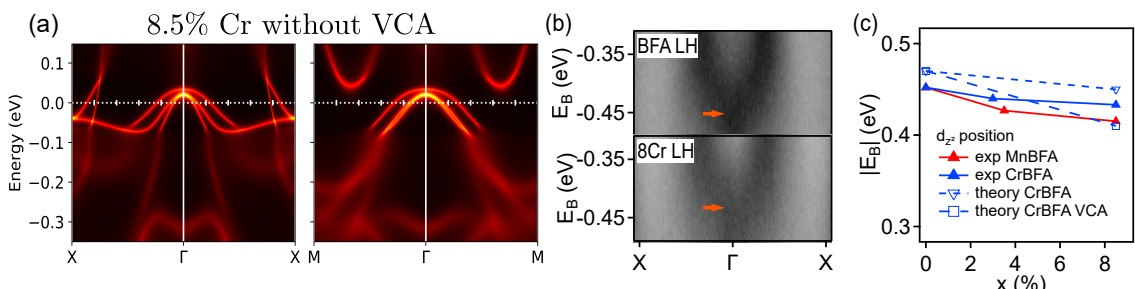

Figure 5: (a) DFT+DMFT spectral functions for the paramagnetic phases at $T = 150$ K for the Cr8.5% material only with the structural change (no VCA). (b) Comparison of higher $E_B$ band energy at $\Gamma$ point for BFA and Cr8.5% samples. (c) Comparison of this energy as a function of dopant concentration with the MnBFA and theoretical values for CrBFA.

In the main text, we showed how the change in $k_F$, and therefore in the hole pockets size, is well reproduced by the VCA, evidencing the role of Cr as a hole dopant. We also noted the change in the hybridization of the $d_{xy}$ derived bands for the Cr8.5% sample.

We now turn attention to electronic states with large $E_B$, to evaluate the validity and limitations of our theoretical approximations. In Fig. 5(b), we compare the bands with $d_{z^2}$ main orbital character [68], at $E_b \approx -0.45$ eV, for the BFA and Cr8.5% samples. The orange arrows show the maximum spectral weight position at the $k_x = 0$ cut. This band position is shifted up with Cr doping, which was also observed for MnBFA as shown in Fig. 5(c), where we also compare it with our theoretical scenarios (dashed lines). The VCA approximation predicts a larger shift than the observed one. Thus, the experimental trend is better described by considering only the structural effects on atomic distances, suggesting that the shift in the $d_z^2$ derived band can be associated with the change in the orbital hybridizations caused by the chemical substitution. Indeed, this effect is more prominent in MnBFA, where charge doping is barely observed [30]. A more qualitative observation is that the bottom of the bands forming the electron pockets is weakly dependent on Cr content, further illustrating that states with larger $E_B$ are less affected by the charge doping and better described without the VCA.

Based on the overall agreement between experiments and our VCA calculations, we can assert that the role of Cr as a hole dopant in CrBFA is transparent for states in the vicinity of the FS and is in sharp contrast to the case of Mn-substituted samples [26,27,30,34]. Nevertheless, the structural changes affecting the orbital hybridization are important to describe the entirety of the electronic structure, affecting mainly the states at larger $E_B$.

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
