# Peer review of "Hole doping and electronic correlations in Cr substituted BaFe$_{2}$As$_{2}$"

_SciPost Physics, doi:SciPost Phys. 17, 141 (2024)_

## Round 1 · Referee Report · Huiqian Luo (Referee 1) · 2024-6-20

Strengths

  1. Full comparison between ARPES experimental results and theory calculations;
  2. High quality data;
  3. Clear conclusions;

Weaknesses

  1. More analysis on the band renormalization will be better

Report

This manuscript reports a combined study on ARPES and band calculations on Cr doped BaFe2As2, a iron-based compound without superconductivity. It is indeed a puzzle that the Cr doping in the 122 familiy of iron-based superconductors. Most of transition metal dopants, such as Ni, Co, Ir, Rh, Pd, Ru, will induce superconductivity in BaFe2As2, except for Cr and Mn, which suppose to give hole dopings in the system. In fact, there is no hole doped on iron site for all superconducting compounds in 122 family. This manuscript give a possible explanation, while the Cr substitution indeed enlarge the hole pockets, which means hole doping into the system, but the electron pockets do not change much. The absence of superconductivity are primarily due to the competition between Cr local moments and the Fe-derived itinerant spin fluctuations. I believe their analysis and conclusions are solid and worthy to publish on SciPost.
I only have one minor suggestions for their consideration: as they have both DFT+DMFT calculation results and ARPES results, is there possible to give the results of orbital- and momentum-dependent band renormalization, namely the ratio between the effective mass and electron mass. To compare the the band renormalization factors upon Cr doping, will certainly give a clear picture about the electronic correlations in this system.

Requested changes

More discussions on the band width and renormalizations.

Recommendation

Publish (easily meets expectations and criteria for this Journal; among top 50%)

  • validity: high
  • significance: high
  • originality: high
  • clarity: high
  • formatting: excellent
  • grammar: excellent

Author:  Marli dos Reis Cantarino  on 2024-08-23  [id 4715]

(in reply to Report 1 by Huiqian Luo on 2024-06-20)
Category:
remark
answer to question

We thank Dr. Huiqian Luo for his report.
From the DFT+DMFT calculations, we can extract the quasiparticle weight (Z) as a function of orbital, directly from the Self-energy, without having to consider parabolic band dispersion approximation. The weight ranges between ~0.3-0.5, indicating a correlated regime, having the tendency to decrease as a function of doping. This is shown in the attached figure for our calculations with VCA as dashed lines. By parabolic band fitting, we can extract the effective mass m* which, in this limited approximation, is inversely proportional to Z. By taking Z = 1/m* (valid for the Fermi liquid regime), we can compare the calculated with experimental Z (solid lines). We can see that the experiments show a considerably heavier band close to the Fermi surface for the outer hole pocket of dxy character. The calculation, although capturing the pocket size trend, did not bring much in terms of orbital dependency results. The main conclusion is related to localization as a function of doping, which we had already seen from the ARPES spectral function analysis. Since the parabolic band dispersion approximation is not so precise, we prefer not to bring this comparison to the manuscript.

To mention these results, we include the following paragraph on the manuscript pg 8:

“Electronic correlations can also be examined by extracting the quasiparticle weight, or renormalization factor, (Z). From our calculations, Z can be extracted from the self-energy without using the parabolic band dispersion approximation. These calculations show Z values between $0.3-0.5$, with Z decreasing (more correlated) with doping. From experiments, the mass renormalization ($m^*$) can be obtained by parabolic fittings of the band dispersions. We observed that near the Fermi surface the outer hole pocket, of $d_{xy}$ orbital character, is much heavier ($m^*/m_e$ = 15) than predicted by theory ($m^*/m_e$ = 2.7) and that the orbital dependence of the renormalization factors is stronger. These results, however, may only indicate that accessing correlations by the parabolic approximation is not the best strategy in the case of Hund’s metal [64].”

Attachment:

---

## Round 1 · Referee Report · Anna Galler (Referee 2) · 2024-7-10

Strengths

  • Extensive and interesting experimental study supported by theoretical calculations.
  • Transparent presentation of the results.

Weaknesses

  • Rather technical and lengthy paper.

Report

The submitted manuscript presents an interesting and comprehensive ARPES study of Cr-substituted BaFe2As2. The findings are underlined by DFT+DMFT computations. I believe it is suitable for publication in SciPost Physics, after the authors address the reviewers' requested changes.

Requested changes

1) The manuscript is rather long and some passages are quite technical. I believe that the paper could profit from a broader readership if the authors find a way to present their most important findings in a more succint way, e.g. by moving some of the technical details to an appendix (e.g. the paragraph and corresponding figure regarding the shifted dz2 band on pg.8, …)

2) Besides the VCA DFT+DMFT computations, the authors also present computational results without VCA (e.g. in figure 1e). If I understand correctly, in these computations only the lattice parameters have been adjusted to the ones of the Cr-doped samples? This might not account for all potential changes in hybridisation effects due to the presence of Cr. I think that showing these results without VCA does not add much to the paper (the spectral function in figure 1e looks essentially the same as in fig. 1d, in fig. 4b the experimental results lie in between the theory results with and without VCA, so I would not conclude much from this), while it may confuse less experienced readers. So I would move these figures together with the corresponding discussions to an appendix, maybe entitled ‘Validity and limitation of the theoretical approach’. In the main text it is sufficient to mention that the VCA approximation does not account for all experimentally observed results, as the authors already do mention on pg. 5.

3) My last concern regards some of the conclusions the authors draw from their study. They state that superconductivity is lacking in CrBFA mainly because of the competition between the Cr local moments and the Fe magnetism (pg. 11). While this sounds plausible, it is not something which is directly investigated in their work, but rather a speculation compatible with their results and should also be formulated in that way. Thus, I suggest that also in the conclusions the authors stick more closely to their results and what they can directly conclude or rule out. Furthermore, I am not sure if the suggested breakdown of low-energy effective models (pg. 11) is a plausible conclusion. Wouldn’t a higher level of theory, which can account e.g. for spin fluctuations and magnetism, or investigations at lower temperature, possibly be able to capture the physics of the material, despite concentrating on the low-energy degrees of freedom?

Recommendation

Ask for minor revision

  • validity: high
  • significance: high
  • originality: good
  • clarity: high
  • formatting: excellent
  • grammar: excellent

Author:  Marli dos Reis Cantarino  on 2024-08-23  [id 4716]

(in reply to Report 2 by Anna Galler on 2024-07-10)

We thank the referee for their comments and suggestions. About the requested changes, we answer in turn:

1 and 2 - regarding the manuscript length and DFT+DMFT calculation details, we followed the referee’s suggestions (also pointed out by ref 3) and moved the VCA and no VCA comparison to the appendix. The dz² band discussion, which relates directly to the approximation limitation, was also moved.

3 - According to the referee’s suggestion, we decided to focus on the aspects that are directly related to our data. Indeed, we do not show a breakdown of low-energy effective models but rather of a particular prediction of the model: i.e. that the magnetism in doped BFA should be connected with the changes in the FS caused by doping. We believe that it is well supported by our experiments that in the case of Cr and Mn-substituted BFA this connection is broken. But it could be indeed the case that a higher order theory can capture the minimal physics.

We have removed the sentence:
“It is thus suggested that low-energy effective models are not adequate to understand the evolution of magnetism for these substitutions.”

We had also rewritten the final paragraph:
“The absence of SC in Cr- and Mn-substituted BaFe$_2$As$_2$ remains a topic of discussion despite intensive research in the past years. In the case of MnBFA, recent experimental work proposes that electronic correlations and the disordered scattering of the Fe-derived magnetic excitations should be part of any minimal model addressing this problem [29, 30]. In the case of CrBFA, the present paper shows that what distinguishes Cr from another hole-doped material, as K-substituted BFA, is indeed the underlying competition between the Cr- and Fe-derived magnetism. This finding suggests that the magnetic scattering of the Fe-derived excitation by Cr suppresses SC in CrBFA.”

---

## Round 1 · Referee Report · Anonymous (Referee 3) · 2024-7-22

Strengths

- First ARPES experiment on Cr-doped BaFe2As2.
- The manuscript gives a good summary of previous literature on Mn and Cr doped iron pnictides, as well as the open issues.

Weaknesses

- Despite the data quality, it is not always easy to understand how the extracted dispersion fit the original data. Maybe the presentation of the data could be clearer.
- The MDC fitting is difficult on these rather broad peaks, which makes the comparaison with theory a bit tentative.

Report

Hole doping via Ba/K substitutions and electron doping via Fe/Co substitutions have revealed high temperature superconductivity in BaFe2As2. These topics have been heavily covered by a number of experiments, including ARPES. The case of Mn or Cr substitutions is less explored and indeed still unclear. Are these substitutions producing doping and, if yes, why are the resulting compounds not superconducting ? To what extent is the localized moment on Mn or Cr interacting with the properties of the compound ? These are the questions adressed in this mansucript by ARPES on the experimental side and LDA+DMFT calculations on the theory side. 1- Case for hole doping This is certainly one of the main argument of the manuscript, but it could be stronger if the dispersion of the hole pockets in Fig. 1 were more clearly shown - and especially their agreement with the markers overlayed on the figures. Maybe zooming around Gamma would be enough to make this clearer. In particular, I cannot distinguish the dxy outer hole pocket at all. Then, In Fig. 1c, there is a dxy hole pocket crossing EF along GX and below EF along GM. This does not make sense to me (it should have the same position at Gamma). This is not commented anywhere in the text, except if this is supposed to be explained by the change of orbital character discussed in 3rd paragraph, page 5, but I do not understand well what the authors mean there. Can they clearly explain if they still observe 3 hole pockets at 8.5% Cr doping and, if yes, which ones they are (why is there a blue hole pocket in LH at roughly the same position than the green hole pocket in LV ?) ? I like the evolution of kF shown in Fig. 3(j-k) and its comparaison with theory, although the way dispersion are extracted for the electron pocket remains unclear to me. Finally, I would appreciate some discussion of why Cr doping in CsFe2As2 was reported in [44] not to produce hole doping. 2- MDC analysis This is clearly a more tentative part from the experimental point of view, especially because Cr is indeed very likely to induce disorder giving a constant term to the linewidth, as rightly acknowledged by the authors on top of page 10. I find that what is deduced from theory or experiment is not sufficiently clear. For example in the abstract, it is written « our results show orbital-specific correlation effects that support the Hund localization scenario ». I guess this can only refer to theory as only dyz is analyzed experimentally, but then the originality of the calculation is maybe not sufficiently underlined.

Requested changes

1- The number of hole bands and their respective orbital character at 8.5% doping should be clarified. I feel this is really necessary before publishing the paper. 2- If possible, enlarge to dispersion of the hole bands in Fig. 1 (or decrease size of markers). This is not clear enough here. 3- I think the paper would be easier to read with a clearer separation between experiment and theory (now Fig. 1 is experiment, but part of its analysis is in Fig. 3. Fig. 2 is theory, etc). 4- Fig. 4(a-b) discuss quite a different point compared to the rest of the figure (and a less important one). It should be separated.

Recommendation

Ask for minor revision

  • validity: ok
  • significance: ok
  • originality: ok
  • clarity: ok
  • formatting: good
  • grammar: good

Author:  Marli dos Reis Cantarino  on 2024-08-23  [id 4717]

(in reply to Report 3 on 2024-07-22)

We thank the referee for their comments and suggestions. Following are our answers:

1 - Case of hole doping
**** Clearer presentation of the data
Figure 1 in the manuscript has been changed.
*** The dxy hole pocket
We have elaborated further on the subject of the assignment of the orbital character of the bands and the mixed character of the dxy and dzy bands in the case of the 8.5%Cr sample (see below).
Indeed, the blue band in GM-LH direction is not the same as the GX-LH. Rather it is the same as the green one in GM-LV. It is noteworthy that the polarization and measurement-geometry selection rules for the GM direction primarily affect the dxy-derived band (see BFA and Cr3.5%). This also indicates mixed orbital character induced by the increasing Cr content, with the weight of the dxy outer hole pocket reduced in this direction and transferred to the GX direction. Previously, we discussed this effect only in the context of the dxy bands in GX. Following the requested change 2 (see below), we modified Figure 1 and the respective discussion to clarify these aspects.
*** No hole doping in Cr-substituted CsFe2As2
In Ref. [44] the authors report an increase in the Sommerfeld coefficient as a function of Cr content in Cr-substituted CsFe2As2. The result is associated with increasing electronic correlations in this system caused by the Cr introduction. The authors also report ARPES data, focusing on the hole pockets around the Gamma point. Unexpectedly, the hole pocket size barely changes with Cr content, a situation that is reminiscent of the much-debated case of Mn-substituted BaFe2As2. Therefore, from the standpoint of the size of the hole pockets, Cr is not doping holes in CsFe2As2.

2 - MDC analysis
*** Abstract and dyz band
The abstract was changed to make it clear that only the spectral features related to the dyz were extracted from experiments. We called attention to this fact later in the text (introduction and conclusion) as well.
We also would like to assert that the analysis of the spectral features of the dyz is a very robust procedure. This is because for measurements along GX and LV polarization, only one hole band is observed close to the Gamma point (see Fig 1). In the other cases, the spectral analysis is indeed “tentative” and thus was not included in the present version of the paper.

In Figure 4 of the ArXiv V01 of this work (https://arxiv.org/pdf/2312.09014v1) this was included. For reference, the Figure 4 is attached. Panels (a) and (b) show this analysis for the BFA sample. In panel (b), two crossing bands are analyzed for GammaM-LH configuration, and the results are satisfactory. In panel (e) we show some details of the fitting for a selected MDC (two bands = four Lorentzians).
But for Cr8.5%, whereas the results for one band are satisfactory (panel (c)), results for two bands (panel (d)) fail to describe some parts of the spectra. In panel (f) this is explicitly shown in the example. We thus excluded this analysis from the present version of the manuscript to focus only on the spectral analysis of the dyz band, which is a trustworthy and robust analysis for all samples.

Regarding the requested changes:

1 - This is now clarified in the main text. On pages 4-5, the discussion was rewritten for clarity and a new paragraph was included to discuss this point further. This discussion is now as below:

“The Cr hole doping effect on the bands forming the hole pockets around $\Gamma$ is visible from direct inspection of the data. Indeed, the hole pocket Fermi vectors, $k_{F}$, are increasing with Cr introduction, in line with the expectation of hole-doping by Cr substitution. Even for the highest Cr doping level in this work ($8.5\%$Cr), a total of three hole pockets are still visible around $\Gamma$, being mainly observed in pairs because of the polarization selection rules.
Closer inspection in the case of the $8.5\%$Cr sample, reveals a change in the polarization selection rules at this substitution level. In the upper left panel of Fig. 1(c), the spectral weight of the outer hole pocket is seen crossing the Fermi surface in an experimental configuration ($\Gamma X$, LH polarization) for which one expects to observe only bands with $d_{xz/yz}$ main orbital character. This is evidence that the main orbital character of the $d_{xy}$ derived bands and their hybridization are significantly affected by Cr, as observed for Co substitution [57]. In addition, comparing the upper and bottom right panels of Fig. 1(c), one observes a relatively weak polarization dependence. The green band in the upper right panel appears where we would expect, based on the other samples, to observe the outer hole pocket. Instead, this band seems related to the band at the bottom panel and thus is of main $d_{yz}$ orbital character. This also evidences the underlying change in the hybridization of the bands with main $d_{xz/yz}$ orbital characters.”

We also changed the color of the band on Fig 1(c) GM-LH panel from blue to green. Since these bands are broad and overlapping, EDC second derivatives were used to extract some of the points, however, this method is not reliable close to the FS, due to the convolution of the lineshape and Fermi-Dirac distribution. Therefore, we removed the questionable points extracted from EDC second derivative from the green (formerly blue) band, since it misrepresents the actual band dispersion. We claim in the new version of the manuscript, as seen above, that these bands are of dyz character, and the dxy contribution that should be expected for GM-LH is present in the form of a new orbital hybridization.

2 - The range of the binding energy was slightly modified to allow a better visualization of the bands, the fitted band points are plotted smaller for clarity.

3 and 4 - Following the referee’s suggestions, accordingly also with ref 2, we moved the VCA and no VCA comparison to the appendix. The dz² band discussion, which relates directly to the approximation limitation, was also moved. The theory left in the main text of the manuscript is only necessary to address the main experimental results discussions directly, namely, the hole pocket sizes and spectral band analysis.

In this regard, Fig 3 presents data analysis that is motivated by the theoretical results presented in Fig 2. We feel that presenting first all data analysis and then the theoretical results would somehow break this interplay between theory/experiment that was indeed part of our process. We rounded the text, making it shorter and more appealing. We invite the referee for a critical reassessment of the text. We hope that the referee will find that the text now reads smoothly.

Attachment:

---

## Round 2 · Referee Report · Anna Galler (Referee 2) · 2024-9-9

Report

The authors have carefully addressed my previous comments, and further improved the manuscript. I think the manuscript is now ready for publication in SciPost Physics.

Recommendation

Publish (meets expectations and criteria for this Journal)

---

## Round 2 · Referee Report · Anonymous (Referee 3) · 2024-9-10

Report

I find that the changes in Fig. 1 and Fig. 4 simplify a bit the reading of the paper and appreciate the change in the abstract about dyz band and orbital selectivity. Otherwise, there are relatively minor changes to the manuscript, the experimental hole bands still cannot be very clearly seen and the discussion of MDC linewidths does not convince me.
However, I find that these are interesting and stimulating findings, worth publishing in SciPost.

Recommendation

Publish (meets expectations and criteria for this Journal)

---

## Round 2 · List of Changes

Figures

Fig. 1: We removed panel (e) from this figure, moving the panel to the new Fig. 05 in Appendix A. The caption was changed accordingly. The range of the binding energy was slightly modified to allow a better visualization of the bands, the fitted band points are plotted smaller for clarity. One band on the GammaM-LH in panel (c) was changed from blue to green. The discussion was updated accordingly.

Fig. 4: We removed panels (a) and (b), moving them to the new Fig. 05 in Appendix A. The caption was changed accordingly.

NEW Fig. 05: This figure includes all the removed panels from Figs. 1 and 4, along with the discussion moved from the main text to the appendix

Abstract

The abstract was changed to make it clear that only the spectral features related to the dyz band were quantitatively experimentally probed. The introduction/conclusion was updated accordingly, softening the mention of correlation effects.
We softened the claim about SC suppression in the abstract.

Discussion

We added a new paragraph on page 9 to address the mass renormalization from theory and experiments.
The discussion on pages 4-5, regarding the band assignment for the 8.5%Cr sample, was changed to explain better the changes introduced by doping and their orbital assignment.
We also improved the description of the number of observed pockets, with a focus on the 8.5%Cr sample.

Conclusion

We modified the conclusions to soften our remarks and focus on the extent of our experimental results. We remove the following sentence:
“It is thus suggested that low-energy effective models are not adequate to understand the evolution of magnetism for these substitutions.”
And we changed the final paragraph.

Appendix

A new appendix section entitled “Note on the theoretical calculations” was added, addressing the more technical concerns in detail without impairing the manuscript's main points and readability. All the discussion related to the dz² band and the different theory approximations to account for the doping were moved to this new section.

---

## Round 3 · List of Changes

Abstract

The abstract was changed. In particular, we now motivate the paper primarily by the behavior of Tsdw, which is directly connected to our data (as suggested by the referees in the previous round). We also emphasize that the DFT+DMFT results support the fractional scaling of the self-energy imaginary part.

Discussion

About the topic of the bands forming hole pockets, selection rules and band main orbital character assignment:
- On page 4, a new paragraph was added and the last paragraph was revised
- On page 5, the first two paragraphs were revised.

About the topic of the spectral analysis:
- On page 8, all four paragraphs introducing/discussing this topic were revised.

Other changes:
- On pages 9/10 we reworded the discussion about the phase diagram for clarity and conciseness (last paragraph before conclusion).

---

## Editorial Decision

published